# MITIGATING THE LIMITATIONS OF MULTIMODAL VAES WITH COORDINATION-BASED APPROACH

## ABSTRACT

One of the key challenges in multimodal variational autoencoders (VAEs) is inferring a joint representation from arbitrary subsets of modalities. The state-of-the-art approach to achieving this is to sub-sample the modality subsets and learn to generate all modalities from them. However, this sub-sampling in the mixture-based approach has been shown to degrade other important features of multimodal VAEs, such as quality of generation, and furthermore, this degradation is theoretically unavoidable. In this study, we focus on another approach to learning the joint representation by bringing unimodal inferences closer to joint inference from all modalities, which does not have the above limitation. Although there have been models that can be categorized under this approach, they were derived from different backgrounds; therefore, the relation and superiority between them were not clear. To take a unified view, we first categorize them as coordination-based multimodal VAEs and show that these can be derived from the same multimodal evidence lower bound (ELBO) and that the difference in their performance is related to whether they are more tightly lower bounded. Next, we point out that these existing coordination-based models perform poorly on cross-modal generation (or cross-coherence) because they do not learn to reconstruct modalities from unimodal inferences. Therefore, we propose a novel coordination-based model that incorporates these unimodal reconstructions, which avoids the limitations of both mixture and coordination-based models. Experiments with diverse and challenging datasets show that the proposed model mitigates the limitations in multimodal VAEs and performs well in both cross-coherence and generation quality.

## 1 INTRODUCTION

Deep generative models have recently shown high performance on multimodal data, including images and captions. In particular, multimodal learning with variational autoencoders (VAEs) (Kingma & Welling, 2013; Rezende et al., 2014) has attracted much attention to obtain a joint representation from modalities (Wu & Goodman, 2018; Shi et al., 2019; Sutter et al., 2021). Such representations can be used for predicting common concepts from these modalities or for generating other modalities.

An essential challenge of multimodal VAEs is inferring a joint representation from subsets of modalities. State-of-the-art models attempt to accomplish this by training to generate all modalities from a joint representation inferred from missing (or sub-sampled) modalities. These are called mixture-based multimodal VAEs (Daunhawer et al., 2021a), and examples include MMVAE (Shi et al., 2019) and MoPoE-VAE (Sutter et al., 2021) [1]. However, it has been pointed out that the quality of modality generation is lower than that of unimodal VAEs, and furthermore, cross-generation between modalities, or cross-coherence, can be degraded. This is an inherent limitation on mixture-based VAEs due to modality sub-sampling during training, and theoretical and empirical evidence shows that this limitation cannot be avoided (Daunhawer et al., 2021a).

To alleviate this issue, we focus on another approach in multimodal VAEs that brings the representation inferred from each modality closer to that inferred from all modalities. This approach avoids the

---

[1] Daunhawer et al. (2021a) also include a special case of MVAE (Wu & Goodman, 2018) in mixture-based models. However, we focus on the mixture-based nature of cross-generation of modalities from subsets as their shortcomings, so we exclude MVAE, which does not perform cross-generation.

limitation of degraded generation quality because its objective does not include generation from sub-sampled modalities. Models that fit this approach have been proposed, but their objectives are derived from different backgrounds. For example, MVTCAE (Hwang et al., 2021), which can be regarded as one of them, is derived its objective from the total correlation; therefore, its relationship to the multimodal VAEs' objective, multimodal evidence lower bound (ELBO), is not clear. Moreover, another model of this approach, MMJSD (Sutter et al., 2020), has been shown experimentally to perform worse than MVTCAE (Hwang et al., 2021), but the theoretical reason for this is unclear.

We first categorize them as *coordination-based multimodal VAEs* [2] and show that these models can be viewed as unified, i.e., all are derived from multimodal ELBO, and that the differences in models correspond to whether they are more tightly lower bounded. We also prove that MVTCAE encourages unimodal posterior to approach the average of joint posterior, which might be one of the reasons for learning good inference. Next, we point out that the existing coordination-based VAEs still have a shortcoming in cross-coherence since no generation from unimodal inferences is included in the training. We then propose a novel objective that introduces unimodal reconstruction terms. Since this objective does not include cross-generation, it also avoids the generation degradation issue of the mixture-based models while mitigating the issue of coordination-based models. Note that our model alleviates the issues by only changing the objective, unlike other methods for improving multimodal VAEs by changing the architecture, such as introducing additional latent variables (Tsai et al., 2018; Hsu & Glass, 2018; Sutter et al., 2020; Palumbo et al., 2022).

We conducted experiments on five diverse multimodal datasets, including challenging ones, e.g., those that cannot be adequately trained with existing models or require additional architecture for adequate training. We confirmed that the proposed method outperforms existing mixture- and coordination-based models in terms of cross-coherence and generation quality.

## 2 MULTIMODAL VAEs

Suppose that we have a set of multimodal examples, where each example is a set of $M$ modalities $X = \{\mathbf{x}_m\}_{m=1}^M$. We assume that these examples are derived from data distribution $p_d(X)$ and that each example $X$ has a corresponding common latent concept $\mathbf{z}$, i.e., a joint representation.

Given a training set $\{X^{(i)}\}_{i=1}^N$, our goals are to infer a joint representation from the subset of modalities $X_S \subseteq X$ [3] and to generate another subset $X_{S'} \subseteq X$ via that representation. In other words, we aim to obtain an inference $p(\mathbf{z}|X)$ and a joint distribution $p(X, \mathbf{z}) = \prod_{\mathbf{x}_m \in X} p(\mathbf{x}_m|\mathbf{z})p(\mathbf{z})$.

To achieve this, it is a natural choice to use VAEs (Kingma & Welling, 2013), deep generative models that can learn inference to acquire representations from data in addition to the generation. The generative model $p_\theta(X|\mathbf{z}) = \prod_{m:\mathbf{x}_m \in X} p_\theta(\mathbf{x}_m|\mathbf{z})$ is parameterized by deep neural networks, and the prior of the latent variable is set as a standard Gaussian $p(\mathbf{z}) = \mathcal{N}(\mathbf{0}, \mathbf{I})$. The objective of VAEs is to maximize the expected log-likelihood over data distribution Since it is tractable to optimize this likelihood directly, we introduce an approximated posterior $q_\phi(\mathbf{z}|X)$ and instead optimize the following evidence lower bound (ELBO) on the expected log-likelihood:

$$\mathcal{L}(\theta, \phi) \equiv \mathbb{E}_{p_d(X)}[\mathbb{E}_{q_\phi(\mathbf{z}|X)}[\log p_\theta(X|\mathbf{z}] - D_{KL}(q_\phi(\mathbf{z}|X)||p(\mathbf{z}))] \leq \mathbb{E}_{p_d(X)}[p_\theta(X)]. \quad (1)$$

In this paper, the model group that maximizes this *multimodal ELBO* is collectively called *multimodal VAEs*. We aim to optimize Eq. 1 to obtain a joint representation and generate modalities from it given a modality subset $X_S$. However, Eq. 1 only includes the inference given all modalities; therefore, it does not learn to infer representations from subsets.

### 2.1 MIXTURE-BASED MULTIMODAL VAEs

An inference from subsets is commonly expressed as a combination of unimodal posteriors from each modality. Product of experts (PoE) (Hinton, 2002; Wu & Goodman, 2018) $q_\phi^{PoE}(\mathbf{z}|X_S) \equiv p(\mathbf{z}) \prod_{m:\mathbf{x}_m \in X_S} q_{\phi_m}(\mathbf{z}|\mathbf{x}_m)$ [4] and mixture of experts (MoE) $q_\phi^{MoE}(\mathbf{z}|X_S) \equiv$

---

[2]The term *coordination* is taken from *coordination representations*, a category of multimodal learning (Baltrušaitis et al., 2018), in which representations inferred from different modalities are learned close together.

[3]We denote a subset of the multimodal set $X$ by $X_S$ respectively, where $S$ represents a subset of $\{1, .., M\}$.

[4]$q_{\phi_m}$ is the unimodal inference of modality $\mathbf{x}_m$.

$\sum_{m:\mathbf{x}_m \in X_S} q_{\phi_m}(\mathbf{z}|\mathbf{x}_m)$ are known as ways to aggregate inferences. Moreover, the following mixture of products of experts (MoPoE) (Sutter et al., 2021) generalizes them:

$$q_\phi^{MoPoE}(\mathbf{z}|X) \equiv \sum_{S:X_S \in \mathcal{P}(X)} \omega_S \left( \prod_{m:\mathbf{x}_m \in X_S} p(\mathbf{z})q_{\phi m}(\mathbf{z}|\mathbf{x}_m) \right) = \sum_{S:X_S \in \mathcal{P}(X)} \omega_S q_{\phi_S}^{PoE}(\mathbf{z}|X_S), \qquad (2)$$

where $\sum_{S:X_S \in \mathcal{P}(X)} \omega_S = 1$, $\omega_S \in [0,1]$, and $\mathcal{P}(X)$ is a powerset of $X$. This MoPoE aggregation is equivalent to PoE when $X_S = X$ and MoE when $|X_S| = 1$ for all $X_S \in \mathcal{P}(X)$.

Substituting the MoPoE-aggregated joint posterior (Eq. 2) into Eq. 1, we obtain the following:

$$\mathcal{L}(\theta, \phi) \geq \mathbb{E}_{p_d(X)} \left[ \sum_{S:X_S \in \mathcal{P}(X)} \omega_S \left[ \mathbb{E}_{q_\phi^{PoE}(\mathbf{z}|X_S)}[\log p_\theta(X|\mathbf{z})] - D_{KL}(q_\phi^{PoE}(\mathbf{z}|X_S)||p(\mathbf{z})) \right] \right] \equiv \mathcal{L}^M(\theta, \phi).$$
$$(3)$$

Multimodal VAEs with Eq. 3 are called *mixture-based multimodal VAEs* (Daunhawer et al., 2021a). MMVAE (Shi et al., 2019) and MoPoE-VAE (Sutter et al., 2021) are members of this family.

Comparing with Eq. 1, we can see that the reconstruction term (the first term in the expectation) of Eq. 1 is designed to generate all modalities from the sub-sampled modality subset. In other words, mixture-based multimodal VAEs encourage cross-generation during training, aiming at inferring a consistent joint representation from subset modalities and generating modalities from it.

However, mixture-based VAEs have a severe issue in generating modalities, which is one of the essential goals of multimodal VAEs. Daunhawer et al. (2021a) show that the following inequality holds between Eq. 3 and the expected marginal log-likelihood:

$$\mathbb{E}_{p_d(X)}[\log p_\theta(X)] \geq \Delta(X) + \mathcal{L}^M(\theta, \phi; X), \qquad (4)$$

where $\Delta(X) \equiv \sum_{S:X_S \in \mathcal{P}(X)} w_S H(X_{\{1,...,M\}\setminus S}|X_S)$ [5] is the discrepancy independent of the learning parameters. This inequality means that multimodal ELBO of mixture-based VAEs cannot be tighter than this discrepancy $\Delta(X)$ no matter how much it is maximized. Therefore, the quality of their generating modalities always degrades compared to unimodal VAEs. Moreover, this discrepancy, by definition, depends on how much different information one modality subset $X_{\{1,...,M\}\setminus S}$ has compared to the rest of them $X_S$. Therefore, the more information specific to each modality, the larger this quantity and the worse the generation quality.

This also leads to degrading cross-coherence (Shi et al., 2019), or the ability to generate the same semantic meaning across modalities conditionally. Intuitively, mixture-based VAEs are required to take sub-sampled subsets as inputs to generate the rest, but these inputs have insufficient information for that, resulting in degraded cross-generation.

## 2.2 Avoiding the limitation of multimodal VAEs

What brings about this irreducible discrepancy $\Delta(X)$? According to the proof by Daunhawer et al. (2021a) (see B.4 in Daunhawer et al. (2021a)), it is due to the inclusion in the objective of a cross-generation term, i.e., a term that generates unobserved modalities from observed modalities. Therefore, one promising way to avoid this limitation is to *not include in the objective of the multimodal VAE the cross-generation term*[6].

Note that MVAE (Wu & Goodman, 2018), which first uses PoE for the joint posterior, uses the sum of ELBOs over all modality subsets as the objective, but does not include cross-generation and only includes *reconstruction* of subsets[7], thus enabling high-quality generation. However, MVAE cannot perform cross-generation between different modalities because its objective does not include terms to encourage it. Despite this significant drawback, determining how to work around the limitation of mixture-based models leads to our proposed model described later.

---

[5] $H(X_{\{1,...,M\}\setminus S}|X_S)$ is the conditional entropy of $X_{\{1,...,M\}\setminus S}$ given $X_S$.

[6] Note that the reconstruction term $\mathbb{E}_{q(\mathbf{z}|X_S)}[\log p(X_{S'}|\mathbf{z})]$ (where $X_{S'} \in X_S$), i.e., all the modalities to be generated are subsets of the input, is not included in the cross-generation term discussed here.

[7] Wu & Goodman (2018) state that MVAE utilizes a sub-sampling training paradigm, but this means "ELBOs with sub-sampled modality sets as input", i.e., each ELBO does not include cross-generation term.

## 3 Coordination-based multimodal VAEs

MVTCAE (Hwang et al., 2021) and MMJSD (Sutter et al., 2020) have been proposed to learn to infer joint representation from subset modalities by bringing joint and unimodal posteriors closer together. We find that they omit sub-sampling in the objective unless the posterior is a mixture distribution and thus can learn the joint representation *without being affected by the limitation of the mixture-based VAEs*. We first categorize them as a unified *coordination-based multimodal VAEs*[8].

### 3.1 MVTCAE and MMJSD

The objectives of MVTCAE and MMJSD are expressed as follows:

$$\mathbb{E}_{p_d(X)}\left[\mathbb{E}_{q_\phi(\mathbf{z}|X)}[\log p_\theta(X|\mathbf{z})] - \sum_{m=0}^{M} \pi_m \mathcal{D}_m\right], \mathcal{D}_m = \begin{cases} D_{KL}(q_\phi(\mathbf{z}|X)||q_{\phi_m}(\mathbf{z}|\mathbf{x}_m)) & \text{(MVTCAE)} \quad (5) \\ D_{KL}(q_{\phi_m}(\mathbf{z}|\mathbf{x}_m)||f_\mathcal{M}(\mathbf{z}|X)) & \text{(MMJSD), (6)} \end{cases}$$

where $\sum_m \pi_m = 1$, $\pi_m \in [0, 1]$, and function $f_\mathcal{M}$ defines the mixture of $\{q_{\phi_m}(\mathbf{z}|\mathbf{x}_m)\}_{m=0}^M$. For simplicity of notation, we use $q_{\phi_0}(\mathbf{z}|\mathbf{x}_0) \equiv p(\mathbf{z})$. We denote the entire objective of MVTCAE and MMJSD as $\mathcal{L}^C(\theta, \phi)$ and $\mathcal{L}^{C'}(\theta, \phi)$, respectively.

MVTCAE is derived as a lower bound for the total correlation[9]. When $M = 2$, this model is equal to JMVAE (Suzuki et al., 2016). JMVAE uses a network for joint posterior separately from unimodal posteriors, while MVTCAE represents joint posteriors by PoE-aggregating unimodal posteriors.

MMJSD is derived by replacing the KL divergence of the multimodal ELBO with the Jensen-Shannon divergence. It is proved (in Lemma 2 in Sutter et al. (2020)) that $\mathcal{L}^{C'}(\theta, \phi)$ is a lower bound for multimodal ELBO with $f_\mathcal{M}(\mathbf{z}|X)$ replaced by the dynamic prior: $p^f(\mathbf{z}|X) = f(\{q_{\phi_m}(\mathbf{z}|\mathbf{x}_m)\}_{m=0}^M)$, where $f$ is a function that encompasses each unimodal posterior, and Sutter et al. (2020) choose the weighted PoE. This replacement can be thought of as introducing the *prior knowledge* that all modalities have the same joint representation (Sutter et al., 2020).

Note that in Sutter et al. (2020), the posterior of MMJSD is assumed to be MoE, which is affected by the limitation of mixture-based VAEs. Since we would like to focus on the properties of MMJSD contributing to its coordination-based nature, we call *MMJSD-PoE* the model that replaces the posterior in Eq. 6 with PoE. Compared to the objective of MVTCAE, the *direction* of KL divergence in Eq. 6 is reversed: MVTCAE uses forward KL divergence, while MMJSD(-PoE) uses reverse KL divergence. Hwang et al. (2021) show that MVTCAE performs better than MMJSD by experiment.

### 3.2 The relation between coordination-based multimodal VAEs

Because MVTCAE and MMJSD were proposed for different purposes, their relationship is unclear. In particular, since MVTCAE was derived from the total correlation, it is unclear whether it belongs to multimodal VAEs (i.e., valid ELBOs). We first show that MVTCAE is multimodal ELBO.

**Lemma 1** *The objective of MVTCAE $\mathcal{L}^C(\theta, \phi)$ is equal to the multimodal ELBO in Eq. 1 with a weighted PoE-based dynamic prior as its prior.*

**Proof 1** *Following MMJSD, we introduce a weighted PoE-based dynamic prior for $\mathcal{L}^C(\theta, \phi)$:*

$$\mathbb{E}_{p_d(X)}[\mathbb{E}_{q_\phi(\mathbf{z}|X)}[\log p_\theta(X|\mathbf{z})] - D_{KL}(q_\phi(\mathbf{z}|X)||p^f(\mathbf{z}|X))]]$$

$$= \mathbb{E}_{p_d(X)}\left[\mathbb{E}_{q_\phi(\mathbf{z}|X)}[\log p_\theta(X|\mathbf{z})] - \mathbb{E}_{q_\phi(\mathbf{z}|X)}\left[\sum_{m=0}^{M} \pi_m \log q_\phi(\mathbf{z}|X) - \pi_m \log q_{\phi_m}(\mathbf{z}|\mathbf{x}_m)\right]\right] = \mathcal{L}^C(\theta, \phi).\square$$

Therefore, MVTCAE can be considered a member of multimodal VAEs. This indicates that the objective of JMVAE is also multimodal ELBO. Note that no assumption is made about the approximate posterior in Lemma 1, and thus holds whether it is aggregated by PoE or MoE. Next, in special cases, the objective of MVTCAE is an upper bound on the objective of MMJSD.

---

[8]The coordination-based VAEs limited to two modalities are not summarized here because they are not scalable. See Section 5 for a description of them.

[9]We changed the coefficients and removed non-optimized terms from the original  (Hwang et al., 2021).

**Lemma 2** *If the posterior is MoE-aggregated, the objective of MVTCAE $\mathcal{L}^C(\theta, \phi)$ is an upper bound on that of MMJSD $\mathcal{L}^{C'}(\theta, \phi)$, i.e., the following inequality holds: $\mathcal{L}^C(\theta, \phi) \geq \mathcal{L}^{C'}(\theta, \phi)$.*

**Proof 2** *Lemma 1 shows that the objective of MVTCAE is equal to a multimodal ELBO with a dynamic prior. In addition, Lemma 2 in Sutter et al. (2020) states that the objective of MMJSD is a lower bound on multimodal ELBO with a dynamic prior and MoE-aggregated posterior. Therefore, the objective of MMJSD is a lower bound on that of MVTCAE if the posterior is MoE-aggregated.* □

Lemma 2 indicates that MVTCAE better approximates the true log-likelihood and thus promotes the good generation, which provides one possible explanation as to why MVTCAE performs better than MMJSD. Note that Lemma 2 comes from the fact that the joint posterior of the KL divergence in MMJSD assumes MoE; therefore, it is still unclear whether this holds when the posterior is PoE.

Furthermore, we show why this direction of KL divergence for MVTCAE is good for multimodal VAEs from a different perspective.

**Lemma 3** *The KL divergence for each $m$ (Eq. 5) brings the average of the joint approximate posterior $q_\phi(\mathbf{z}|\mathbf{x}_m) = \int q_\phi(\mathbf{z}|X)p_d(X_{C/m}|\mathbf{x}_m)dX_{C/m}$ closer to the unimodal posterior $q_{\phi_m}(\mathbf{z}|\mathbf{x}_m)$.*

**Proof 3** *Eq. 5 is transformed as follows[10]:*

$$\mathbb{E}_{p_d(X)}\left[D_{KL}(q_\phi(\mathbf{z}|X)||q_{\phi_m}(\mathbf{z}|\mathbf{x}_m))\right] = I_\phi(X_{C/m}; \mathbf{z}|\mathbf{x}_m) + \mathbb{E}_{p_d(\mathbf{x}_m)}\left[D_{KL}[q_\phi(\mathbf{z}|\mathbf{x}_m)||q_{\phi_m}(\mathbf{z}|\mathbf{x}_m))]\right], \quad (7)$$

*where $I_\phi(X_{C/m}; \mathbf{z}|\mathbf{x}_m)$ is a conditional mutual information and holds greater than 0. Therefore, maximizing the negative KL term (Eq. 5) leads to minimizing the second term in Eq. 7.* □

Lemma 3 holds for any approximate posterior. Therefore, another possible reason MVTCAE is good as multimodal VAEs is that maximizing Eq. 5 explicitly brings the unimodal posterior closer to the joint posterior with the unimodal input.

If the joint inference is PoE-aggregation with Gaussian, the approximate posterior of any input can be computed analytically. Therefore, the second term in Eq. 7 can be set to zero by using PoE-aggregate posterior $q_\phi(\mathbf{z}|\mathbf{x}_m) = p(\mathbf{z})q_{\phi_m}(\mathbf{z}|\mathbf{x}_m)$ instead of unimodal posterior $q_{\phi_m}(\mathbf{z}|\mathbf{x}_m)$. From now on, we use the PoE-aggregate posterior.

## 4   COORDINATED AND UNIMODAL RECONSTRUCTED MULTIMODAL VAE

If the inference can be properly learned, the inference from an arbitrary set of input $X_S$ can be computed analytically as $q_\phi(\mathbf{z}|X_S) = p(\mathbf{z})\prod_{m:\mathbf{x}_m \in X_S} q_{\phi_m}(\mathbf{z}|\mathbf{x}_m)$ due to the nature of PoE. However, $\mathcal{L}^C(\theta, \phi)$ only includes reconstruction from all modalities and does not include generation from arbitrary modalities; thus, the performance of cross-modal generation might be degraded. On the other hand, cross-generation in the objective conversely promotes generation degradation due to the limitation of mixture-based models.

We propose to include the term for *unimodal reconstruction* in the objective to *experience* the generation from unimodal posterior during learning while avoiding these dilemmas. The reconstruction term in Eq. 5 can be rewritten as

$$\mathbb{E}_{p_d(X)}\left[\mathbb{E}_{q_\phi(\mathbf{z}|X)}[\log p_\theta(X|\mathbf{z})]\right] = \frac{1}{2}\mathbb{E}_{p_d(X)}\left[\mathbb{E}_{q_\phi(\mathbf{z}|X)}[\log p_\theta(X|\mathbf{z})]\right] + \frac{1}{2}\sum_{m=1}^{M}\mathbb{E}_{p_d(\mathbf{x}_m)}\left[\mathbb{E}_{q_\phi(\mathbf{z}|\mathbf{x}_m)}[\log p_\theta(\mathbf{x}_m|\mathbf{z})]\right],$$
$$(8)$$

where all inferences in Eq. 8 are made using PoE-aggregated posterior.

This rewritten objective can be converted to include all-modality and unimodal reconstruction, which mitigates the shortcoming of coordination-based VAEs while avoiding the limitation of mixture-based VAEs. We call this *Coordinated and unimodal Reconstructed Multimodal VAE (CRMVAE)*.

The expected value for $\mathbf{x}_m$ of the sum of the second term in Eq. 8 can also be transformed into a reconstruction that includes another input as follows:

$$\mathbb{E}_{p_d(\mathbf{x}_m)}\left[\mathbb{E}_{q_\phi(\mathbf{z}|\mathbf{x}_m)}[\log p_\theta(\mathbf{x}_m|\mathbf{z})]\right] = \mathbb{E}_{p_d(\mathbf{x}_m, \mathbf{x}_{m'})}\left[\mathbb{E}_{q_\phi(\mathbf{z}|\mathbf{x}_m, \mathbf{x}_{m'})}[\log p_\theta(\mathbf{x}_m|\mathbf{z})]\right]. \quad (9)$$

---

[10]This is an application of the derivation used for one or two modality inputs in (Hoffman & Johnson, 2016; Vedantam et al., 2018) to multimodal inputs.

Therefore, optimizing the unimodal reconstruction for each modality leads to the possibility of reconstruction from any modality. This performance can be generalized beyond reconstruction to improve the performance of cross-modal generation.

Note that theoretically, the same is true for the left-hand side of Eq. 8, i.e., from the nature of PoE, it should be possible to learn inferences from any modality simply by optimizing the left-hand side of Eq. 8. However, some inferences might be ignored in actual optimization. In other words, it is true that if each inference is adequately learned, the joint inference will be adequate, but the converse is not necessarily true. Our proposal is empirical and attempts to alleviate this problem of being ignored by having each modality experience inference and generation. This simple method is effective for cross-modal generation in a coordination-based setting, as we will show later in the results.

In the experiment, we set the coefficient in Eq. 8 from $\frac{1}{2}$ to $\pi_m$[11] to prevent the expected value term from increasing when the modality increases and having too large an effect on the overall objective.

## 5 RELATED WORKS

There have been many studies of multimodal VAEs (Suzuki & Matsuo, 2022), and the early ones before mixture-based required additional inference models. Suzuki et al. (2016) proposed to approximate the unimodal inference models to explicitly approach the joint posterior of VAEs, called JMVAE. Vedantam et al. (2018) introduced TELBO, where the objective is composed of the sum of ELBOs that take each set of modalities as input. M$^2$VAE (Korthals et al., 2019) can be regarded as a combination of JMVAE and TELBO, and VAEVAE (Wu & Goodman, 2019) excludes a KL divergence term between the joint inference and prior from M$^2$VAE. Except for TELBO, they can be regarded as precursors to the coordination-based approach; however, the number of inference models needed for these models grows exponentially with the number of modalities.

One important research direction in multimodal VAEs is the introduction of additional latent variables, which includes hierarchical latent variables (Sutter & Vogt, 2021; Wolff et al., 2021; Vasco et al., 2022) and modality-specific latent variables (Tsai et al., 2018; Hsu & Glass, 2018; Sutter et al., 2020; Lee & Pavlovic, 2020; Daunhawer et al., 2021b; Palumbo et al., 2022). This paper aims to improve the performance of multimodal VAEs in a different direction from them, i.e., whether limitations of existing mixture-based models can be mitigated by changing the objective while *maintaining the same architecture*. Note that our proposed model and the introduction of additional latent variables are orthogonal concepts and can be easily incorporated into the proposed model.

## 6 EXPERIMENTS

We use the following five diverse datasets: **MNIST-SVHN-Text** (Sutter et al., 2020), **PolyM-NIST** (Sutter et al., 2021), **Translated-PolyMNIST** (Daunhawer et al., 2021a), **CUB** Wah et al. (2011), and **Bimodal CelebA** (Sutter et al., 2020). For a more description of the dataset, see the appendix. These are commonly used in studies of multimodal VAEs, but to our knowledge, this is the first study in that experiments have been conducted on *all* of these datasets. In particular, CUB and Translated-PolyMNIST were noted to be challenging datasets that could not be successfully trained in existing models (Daunhawer et al., 2021a). It also shows that Bimodal CelebA and CUB could only be learned in models with additional latent variables (Sutter et al., 2020; Palumbo et al., 2022).

For models trained in all modalities, we evaluate both 1) cross-coherence and 2) generation quality given arbitrary modalities. We measure each of them in terms of 1) the test accuracy of a pre-trained classifier[12] that predicts labels from cross-generated images and 2) the Fréchet inception distance (FID) (Heusel et al., 2017) in cross-generated modalities. For MNIST-SVHN-Text, joint coherence is also verified by generating random images in all modalities and evaluating whether all classes predicted by them are consistent. Note that we evaluated the generated results of the text modality in MNIST-SVHN-Text in terms of the Levenshtein distance from the true text (without space characters) since we found that the pre-trained classifier from Sutter et al. (2021) did not correctly evaluate the generated results (we discuss in the experimental results.) To assess the cross-coherence for CUB, we

---

[11]This leads to a change in the balance between the reconstruction and KL terms compared to other models as the number of modalities increases. This will be examined in Appendix H.

[12]We used the pre-trained classifier in PolyMNIST and MNIST-SVHN-Text from Sutter et al. (2021).

Table 1: Cross-coherence on MNIST-SVHN-Text. M, S, and T represent MNIST, SVHN, and Text. The top row shows the modalities generated, and the second row shows modalities conditioned when generating them. For example, the second column of the MVAE results (accuracy 0.30) represents the cross-generation from S to M. A+B means that the modality in the row above is generated from the modalities A and B. R represents the joint coherence of all modalities in a random generation.

| | MODEL | R | M | | | S | | | T | | |
| | | | S | T | S+T | M | T | M+T | M | S | M+S |
|---|---|---|---|---|---|---|---|---|---|---|---|
| | MVAE | 0.04 | 0.30 | 0.11 | 0.30 | 0.36 | 0.29 | 0.67 | 0.13 | 0.20 | 0.22 |
| Mixture | MMVAE | 0.12 | 0.81 | **0.99** | 0.91 | 0.39 | 0.44 | 0.41 | 0.48 | 0.40 | 0.44 |
| | MOPOE-VAE | 0.12 | **0.82** | **0.99** | 0.94 | 0.35 | 0.37 | 0.36 | 0.49 | 0.42 | 0.51 |
| | MMJSD | 0.09 | 0.81 | **0.99** | 0.90 | 0.39 | 0.41 | 0.40 | 0.44 | 0.37 | 0.40 |
| Coodination | MMJSD-POE | **0.15** | 0.12 | 0.78 | 0.80 | 0.55 | 0.73 | 0.86 | 0.71 | 0.14 | 0.74 |
| | MVTCAE | 0.09 | 0.23 | 0.86 | 0.90 | 0.59 | 0.78 | 0.80 | 0.76 | 0.27 | 0.79 |
| | **CRMVAE** | **0.15** | 0.60 | **0.97** | **0.97** | **0.73** | **0.87** | **0.87** | **0.91** | **0.71** | **0.95** |

Table 2: FID scores for cross-generation. Lower is better.

| MODEL | S→M | M→S |
|---|---|---|
| MVAE | 39.0 | 36.0 |
| MMVAE | 91.8 | 307.9 |
| MOPOE-VAE | 85.3 | 312.8 |
| MMJSD | 86.7 | 287.0 |
| MMJSD-POE | 70.0 | 141.4 |
| MVTCAE | 41.6 | **31.8** |
| **CRMVAE** | **24.1** | 37.3 |

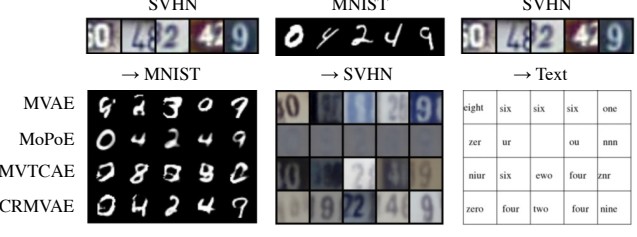

Figure 1: Visualization of cross-generation.

use the cosine similarity between the latent space of the two inferred modalities since there are no labels. We also used this evaluation for Bimodal CelebA.

For the network architectures, we follow Sutter et al. (2020); Daunhawer et al. (2021a), with 64-dimensional latent variables for Bimodal CelebA and CUB, 32 for MNIST-SVHN-Text, and 512 for PolyMNIST and Translated-PolyMNIST. The distributions for image modalities are set to Laplace, and text is set to categorical. We use Adam (Kingma & Ba, 2014) for optimization with a learning rate of 0.0005 and a mini-batch size of 256. We set $\beta$ (the coefficient of the KL divergence term in the multimodal ELBO) to 1 for PolyMNIST and MNIST-SVHN-Text and 0.1 for Translated-PolyMNIST, $\pi_m$ to $\frac{1}{M+1}$, and $\omega_S$ to $\frac{1}{2^M}$. We train 100 epochs for MNIST-SVHN-Text, 150 for CUB, 200 for Bimodal CelebA, and 500 for PolyMNIST and Translated-PolyMNIST. All experimental results are the average of three runs with different seeds. We implement them using PyTorch (Paszke et al., 2019). Our experiments were performed using two machines with five NVIDIA Titan RTX GPUs. We compare CRMVAE with MVAE, mixture-based VAEs (MMVAE, MoPoE, and MMJSD), and coordination-based VAEs (MVTCAE and MMJSD-PoE), all with the same architecture settings.

## 6.1 MNIST-SVHN-TEXT

Table 1 shows the joint and cross-coherence results, and Table 2 shows the FID scores of images on MNIST-SVHN-Text. First, we can confirm that MVAE has high generation quality but the lowest cross-coherence since it learns only the modality reconstruction. We next find that mixture-based models perform poorly in cross-generation of SVHN and Text[13]. According to the qualitative results shown in Figure 1, the cross-generation of Text on MoPoE-VAE is unsuccessful. While Table 1 shows that cross-generation of MNIST appears to perform well, the qualitative results show that the generated images are average with no variation. Table 2 indicates that FID scores of mixture-based models on MNIST are significantly lower than these of MVAE, which corresponds to these results.

Existing coordination-based models mitigate this limitation and perform better on SVHN. Furthermore, FID scores show that the coordination-based models significantly outperform the mixture-based models in terms of generating quality and are comparable to MVAE. Comparing MMJSD-PoE and MVTCAE, MVTCAE performs better in terms of both cross-coherence and generation qual-

---

[13]Note that the results of Text differ significantly from the results of previous studies because the Levenshtein distance is used for the generated results, not for the pre-trained classifiers. We find that the evaluation using the Levenshtein distance is consistent with the qualitative results.

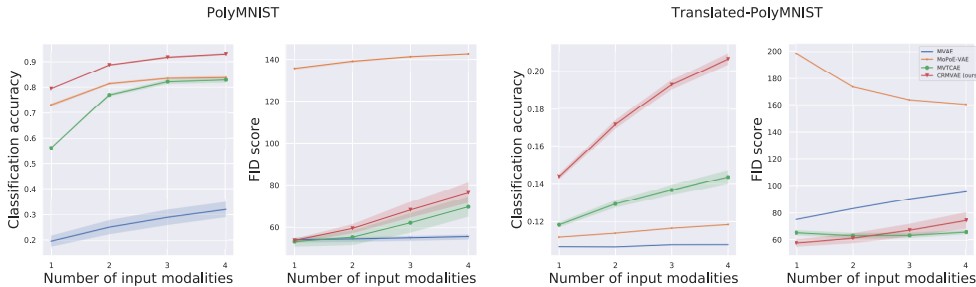

Figure 2: Performance evaluation on PolyMNIST and Translated-PolyMNIST. The left panel for each dataset shows the cross-coherence given subset of size corresponding to each number and the right panel shows the FID score given subset (lower is better). The markers are the means of the three runs, and the error bands are their standard deviations.

ity, confirming the effectiveness of MVTCAE experimentally. However, the performance of the coordination-based is degraded, especially in cross-generation from SVHN.

CRMVAE mitigates coordination-based issues and significantly improves cross-coherence performance. In particular, CRMVAE outperforms all existing methods, including mixture-based methods, for cross-coherence of SVHN and Text. We also confirm that CRMVAE retains the advantage of the coordination-based models, i.e., high-generation quality. In particular, we have confirmed the highest quality of cross-generation on MNIST among all models that include MVAE. The visualization results also confirm that CRMVAE achieves high coherence while maintaining high quality.

The performance of the joint coherence results with random images is poor compared to other studies, which might be due in part to the different methods for evaluating text. The performance of the coordination-based method is relatively high but not significant. The relationship between joint coherence and coordination-based should be studied further.

## 6.2 POLYMNIST AND TRANSLATED-POLYMNIST

Figure 2 shows the results on PolyMNIST and Translated-PolyMNIST. Based on the above results, we select MoPoE-VAE and MVTCAE as mixture-based and coordination-based models for comparison. First the results of the existing models on PolyMNIST show the same trend as on MNIST-SVHN-Text, i.e., there is a trade-off between cross-coherence and generation quality. CRMVAE mitigates these limitations and achieves the highest cross-coherence while maintaining high generation quality. Note that it might seem strange that both MVTCAE and CRMVAE produce lower quality when the number of modalities increases. This is thought to be due to a decrease in the diversity of the generated data distribution as the number of modalities increases.

Translated-PolyMNIST is a slightly modified PolyMNIST, but Daunhawer et al. (2021a) point out that the generation quality and cross-coherence performance of mixed-based models is significantly degraded on this dataset. Our experiments also confirm that most existing models fail to cross-generate across modalities. CRMVAE, on the other hand, clearly improves cross-coherence, although not as much as PolyMNIST. Next, CRMVAE has the highest generation quality, significantly outperforming the mixture-based models. See Appendix I for visualization of the generated images.

## 6.3 CUB AND BIMODAL CELEBA

Figures 3 and 4 show the quantitative and qualitative results for CUB and Bimodal CelebA. We examine the generation quality and cross-coherence performance as an ablation, when varying $\beta$. First, the FID score indicates that MVAEs have high generation quality, while MoPoE-VAEs have significantly lower quality. In most cases, the FID score does not change much or worsen when $\beta$ is increased, while the FID score for MoPoE-VAE improves slightly. This is thought to be because increasing $\beta$ in MoPoE-VAE corresponds to making the distribution closer to the standard Gaussian, which diminishes the effect of generation from sub-sampling. Furthermore, the performance of the coordination-based methods, including CRMVAE, is close to that of MVAE, indicating that they do not have a generation quality issue on these challenging datasets.

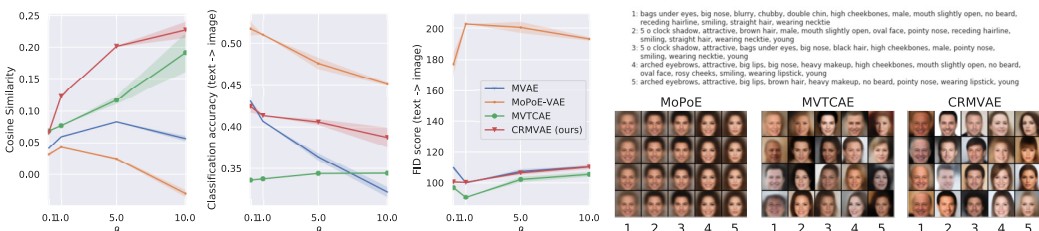

Figure 3: Left: Performance of the cosine similarity of the two modalities in latent space (higher is better, left) and the FID score of the image generated from text (right) for varying $\beta$ in the CUB dataset. Right: Visualization of the CUB images generated by each model ($\beta = 10$, bottom) conditioned on each text (top). The number in each column of each model represents the number of the text being conditioned. Different rows correspond to different sample results.

Figure 4: Left: Performance of the cosine similarity (left), classification accuracy (center), and FID score (right) for varying $\beta$ in the Bimodal CelebA dataset. Right: Visualization of the Bimodal CelebA images generated by each model ($\beta = 5$, bottom) conditioned on each text (top).

Next, the evaluation of cosine similarity shows that MVAE and MoPoE-VAE do not change that much when $\beta$ is increased, while the coordination-based models, including CRMVAE, improve when $\beta$ is increased. This is because, in the coordination-based models, the term adjusted by $\beta$ corresponds to the term that brings the inference of all modalities closer to that of each modality. Compared to MVTCAE, CRMVAE performs better for Bimodal CelebA for all values of $\beta$ and for CUB when $\beta$ is large. Furthermore, the cross-coherence with pre-trained classifiers in Bimodal CelebA also shows that CRMVAE performs better than MVTCAE. Note that MoPoE-VAE has the highest cross-coherence value, but as the qualitative results show, this only improves classification accuracy at the expense of image quality, e.g., diversity (as is the case for MNIST in MNIST-SVHN-Text).

Qualitative results showed the following limitations: MVAE failed to produce images corresponding to the text, and MoPoE-VAE produced only average or blurred images. Comparing MVTCAE and CRMVAE, CRMVAE produces more appropriately conditioned images. For example, in Bimodal CelebA, Text 3 contains a male attribute and CRMVAE produces a corresponding face image, while MVTCAE produces some female face images.

## 7 CONCLUSION AND DISCUSSION

We focused on coordination-based multimodal VAEs to avoid the limitations of multimodal VAEs. We first discuss the new relation among coordination-based multimodal VAEs, point out the issue of unimodal inference not learning from reconstruction in them, and then introduce a new objective involving unimodal reconstruction. Experiments show that the proposed model mitigates existing models' limitations (both mixture-based and coordination-based) and shows overall superior performance on diverse datasets for generation quality and cross-coherence.

One limitation of CRMVAE is that there still remains a trade-off between generation quality and cross-coherence. For example, in the PolyMNIST experiment, CRMVAE performed best in cross-coherence, but the FID score was worse than MVTCAE. The performance of joint coherence was also confirmed to be limited. In addition, although we used several diverse datasets to evaluate performance, these datasets are much smaller than the various multimodal information obtained from the real world. Therefore, we should validate our model using such large multimodal information.

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

## A  DATASET

**MNIST-SVHN-Text** (Sutter et al., 2020) is the MNIST-SVHN (Shi et al., 2019) with the addition of the text modality, which represents the English name of each number as text and has diversity. Different modalities of the same example have the same digits.

**PolyMNIST** (Sutter et al., 2021) consists of a set of five different MNIST images with different backgrounds and handwriting styles, each of which is considered a different modality.

**Translated-PolyMNIST** (Daunhawer et al., 2021a) is composed of the same backgrounds as PolyMNIST with the numbers downscaled[14] and randomly transformed into position in the images. This is intended to allow for more significant variation among modalities than PolyMNIST.

Caltech Birds (**CUB**) Wah et al. (2011) is a dataset with bird images and their captions. We follow Daunhawer et al. (2021a), i.e., we treat as $64 \times 64$ real images without feature extraction.

**Bimodal CelebA** (Sutter et al., 2020) is a dataset of face images from the CelebA (Liu et al., 2015) with additional text describing the corresponding attribute labels and changing the starting index to give a diversity.

---

[14]In setting of this study, the handwritten numbers were downscaled to 70% of their original size.

## B  DETAILED DERIVATION OF PROOF 1

$$
\mathbb{E}_{p_d(X)}[\mathbb{E}_{q_\phi(\mathbf{z}|X)}[\log p_\theta(X|\mathbf{z})] - D_{KL}(q_\phi(\mathbf{z}|X)||p^f(\mathbf{z}|X))]]
$$

$$
= \mathbb{E}_{p_d(X)}\left[\mathbb{E}_{q_\phi(\mathbf{z}|X)}[\log p_\theta(X|\mathbf{z})] - \mathbb{E}_{q_\phi(\mathbf{z}|X)}[\log q_\phi(\mathbf{z}|X) - \log p^f(\mathbf{z}|X)]\right]
$$

$$
= \mathbb{E}_{p_d(X)}\left[\mathbb{E}_{q_\phi(\mathbf{z}|X)}[\log p_\theta(X|\mathbf{z})] - \mathbb{E}_{q_\phi(\mathbf{z}|X)}[\log q_\phi(\mathbf{z}|X) - \sum_{m=0}^{M} \pi_m \log q_{\phi_m}(\mathbf{z}|\mathbf{x}_m)]\right]
$$

$$
\left(\because p^f(\mathbf{z}|X) = \sum_{m=0}^{M} \pi_m \log q_{\phi_m}(\mathbf{z}|\mathbf{x}_m)\right)
$$

$$
= \mathbb{E}_{p_d(X)}\left[\mathbb{E}_{q_\phi(\mathbf{z}|X)}[\log p_\theta(X|\mathbf{z})] - \mathbb{E}_{q_\phi(\mathbf{z}|X)}\left[\sum_{m=0}^{M} \pi_m \log q_\phi(\mathbf{z}|X) - \sum_{m=0}^{M} \pi_m \log q_{\phi_m}(\mathbf{z}|\mathbf{x}_m)\right]\right]
$$

$$
\left(\because \log q_\phi(\mathbf{z}|X) = \sum_{m=0}^{M} \pi_m \log q_\phi(\mathbf{z}|X) \text{ and } \sum_{m=0}^{M} \pi_m = 1\right)
$$

$$
= \mathbb{E}_{p_d(X)}\left[\mathbb{E}_{q_\phi(\mathbf{z}|X)}[\log p_\theta(X|\mathbf{z})] - \sum_{m=0}^{M} \pi_m \mathbb{E}_{q_\phi(\mathbf{z}|X)}[\log q_\phi(\mathbf{z}|X) - \log q_{\phi_m}(\mathbf{z}|\mathbf{x}_m)]\right]
$$

$$
= \mathbb{E}_{p_d(X)}\left[\mathbb{E}_{q_\phi(\mathbf{z}|X)}[\log p_\theta(X|\mathbf{z})] - \sum_{m=0}^{M} \pi_m D_{KL}(q_\phi(\mathbf{z}|X)||q_{\phi_m}(\mathbf{z}|\mathbf{x}_m))\right]
$$

$$
= \mathcal{L}^C(\theta, \phi)\ \square
$$

## C  DETAILED DERIVATION OF PROOF 3

$$
\mathbb{E}_{p_d(X)}\left[D_{KL}(q_\phi(\mathbf{z}|X)||q_{\phi_m}(\mathbf{z}|\mathbf{x}_m))\right]
$$

$$
= \int p_d(\mathbf{x}_m) \int \int q_\phi(\mathbf{z}|X)p_d(X_{C/m}|\mathbf{x}_m) \log \frac{q_\phi(\mathbf{z}|X)}{q_{\phi_m}(\mathbf{z}|\mathbf{x}_m)} dX_{C/m} d\mathbf{z} d\mathbf{x}_m
$$

$$
= \int p_d(\mathbf{x}_m) \int \int q_\phi(\mathbf{z}|X)p_d(X_{C/m}|\mathbf{x}_m) \log \frac{q_\phi(\mathbf{z}|X)p_d(X_{C/m}|\mathbf{x}_m)}{q_{\phi_m}(\mathbf{z}|\mathbf{x}_m)p_d(X_{C/m}|\mathbf{x}_m)} dX_{C/m} d\mathbf{z} d\mathbf{x}_m
$$

$$
= \int p_d(\mathbf{x}_m) \int \int q_\phi(\mathbf{z}|X)p_d(X_{C/m}|\mathbf{x}_m) \log \frac{q_\phi(\mathbf{z}|X)p_d(X_{C/m}|\mathbf{x}_m)}{q_\phi(\mathbf{z}|\mathbf{x}_m)p_d(X_{C/m}|\mathbf{x}_m)} dX_{C/m} d\mathbf{z} d\mathbf{x}_m
$$

$$
+ \int p_d(\mathbf{x}_m) \int \int q_\phi(\mathbf{z}|X)p_d(X_{C/m}|\mathbf{x}_m) dX_{C/m} \log \frac{q_\phi(\mathbf{z}|\mathbf{x}_m)}{q_{\phi_m}(\mathbf{z}|\mathbf{x}_m)} d\mathbf{z} d\mathbf{x}_m
$$

$$
= \int p_d(\mathbf{x}_m) \int \int q_\phi(\mathbf{z}, X_{C/m}|\mathbf{x}_m) \log \frac{q_\phi(\mathbf{z}, X_{C/m}|\mathbf{x}_m)}{q_\phi(\mathbf{z}|\mathbf{x}_m)p_d(X_{C/m}|\mathbf{x}_m)} dX_{C/m} d\mathbf{z} d\mathbf{x}_m
$$

$$
+ \int p_d(\mathbf{x}_m) \int \int q_\phi(\mathbf{z}|\mathbf{x}_m) \log \frac{q_\phi(\mathbf{z}|\mathbf{x}_m)}{q_{\phi_m}(\mathbf{z}|\mathbf{x}_m)} d\mathbf{z} d\mathbf{x}_m
$$

$$
= I_\phi(X_{C/m}; \mathbf{z}|\mathbf{x}_m) + \mathbb{E}_{p_d(\mathbf{x}_m)}\left[D_{KL}[q_\phi(\mathbf{z}|\mathbf{x}_m)||q_{\phi_m}(\mathbf{z}|\mathbf{x}_m))]\right].
$$

## D   DETAILED DERIVATION OF EQ. 7

$$\mathbb{E}_{p_d(X)}\left[\mathbb{E}_{q_\phi(\mathbf{z}|X)}[\log p_\theta(X|\mathbf{z})]\right]$$

$$= \mathbb{E}_{p_d(X)}\left[\frac{1}{2}\mathbb{E}_{q_\phi(\mathbf{z}|X)}[\log p_\theta(X|\mathbf{z})] + \frac{1}{2}\mathbb{E}_{q_\phi(\mathbf{z}|X)}[\log p_\theta(X|\mathbf{z})]\right]$$

$$= \mathbb{E}_{p_d(X)}\left[\frac{1}{2}\mathbb{E}_{q_\phi(\mathbf{z}|X)}[\log p_\theta(X|\mathbf{z})] + \frac{1}{2}\mathbb{E}_{q_\phi(\mathbf{z}|X)}\left[\sum_{m=1}^{M}\log p_\theta(\mathbf{x}_m|\mathbf{z})\right]\right]$$

$$= \mathbb{E}_{p_d(X)}\left[\frac{1}{2}\mathbb{E}_{q_\phi(\mathbf{z}|X)}[\log p_\theta(X|\mathbf{z})]\right] + \frac{1}{2}\int\int q_\phi(\mathbf{z}|X)p_d(X)\sum_{m=1}^{M}\log p_\theta(\mathbf{x}_m|\mathbf{z})d\mathbf{z}dX$$

$$= \mathbb{E}_{p_d(X)}\left[\frac{1}{2}\mathbb{E}_{q_\phi(\mathbf{z}|X)}[\log p_\theta(X|\mathbf{z})]\right] + \frac{1}{2}\sum_{m=1}^{M}\int\int q_\phi(\mathbf{z}|X)p_d(X)\log p_\theta(\mathbf{x}_m|\mathbf{z})d\mathbf{z}dX$$

$$= \mathbb{E}_{p_d(X)}\left[\frac{1}{2}\mathbb{E}_{q_\phi(\mathbf{z}|X)}[\log p_\theta(X|\mathbf{z})]\right] + \frac{1}{2}\sum_{m=1}^{M}\int\int q_\phi(\mathbf{z}|X)p_d(X_{C/m}|\mathbf{x}_m)p_d(\mathbf{x}_m)\log p_\theta(\mathbf{x}_m|\mathbf{z})d\mathbf{z}dX$$

$$= \mathbb{E}_{p_d(X)}\left[\frac{1}{2}\mathbb{E}_{q_\phi(\mathbf{z}|X)}[\log p_\theta(X|\mathbf{z})]\right] + \frac{1}{2}\sum_{m=1}^{M}\int\int\int q_\phi(\mathbf{z}|X)p_d(X_{C/m}|\mathbf{x}_m)dX_{C/m}p_d(\mathbf{x}_m)\log p_\theta(\mathbf{x}_m|\mathbf{z})d\mathbf{z}d\mathbf{x}_m$$

$$= \mathbb{E}_{p_d(X)}\left[\frac{1}{2}\mathbb{E}_{q_\phi(\mathbf{z}|X)}[\log p_\theta(X|\mathbf{z})]\right] + \frac{1}{2}\sum_{m=1}^{M}\int\int q_\phi(\mathbf{z}|\mathbf{x}_m)p_d(\mathbf{x}_m)\log p_\theta(\mathbf{x}_m|\mathbf{z})d\mathbf{z}d\mathbf{x}_m$$

$$\left(\because \int q_\phi(\mathbf{z}|X)p_d(X_{C/m}|\mathbf{x}_m)dX_{C/m} = q_\phi(\mathbf{z}|\mathbf{x}_m)\right)$$

$$= \frac{1}{2}\mathbb{E}_{p_d(X)}\left[\mathbb{E}_{q_\phi(\mathbf{z}|X)}[\log p_\theta(X|\mathbf{z})]\right] + \frac{1}{2}\sum_{m=1}^{M}\mathbb{E}_{p_d(\mathbf{x}_m)}\left[\mathbb{E}_{q_\phi(\mathbf{z}|\mathbf{x}_m)}[\log p_\theta(\mathbf{x}_m|\mathbf{z})]\right].$$

## E   DETAILED DERIVATION OF EQ. 9

$$\mathbb{E}_{p_d(\mathbf{x}_m)}\left[\mathbb{E}_{q_\phi(\mathbf{z}|\mathbf{x}_m)}[\log p_\theta(\mathbf{x}_m|\mathbf{z})]\right]$$

$$= \int\int q_\phi(\mathbf{z}|X)p_d(X)\log p_\theta(\mathbf{x}_m|\mathbf{z})d\mathbf{z}dX$$

$$= \int\int q_\phi(\mathbf{z}|X)p_d(X_{C/m,m'}|\mathbf{x}_m,\mathbf{x}_{m'})dX_{C/m,m'}p_d(\mathbf{x}_m,\mathbf{x}_{m'})\log p_\theta(\mathbf{x}_m|\mathbf{z})d\mathbf{z}d\mathbf{x}_md\mathbf{x}_{m'}$$

$$= \int\int q_\phi(\mathbf{z}|\mathbf{x}_m,\mathbf{x}_{m'})p_d(\mathbf{x}_m,\mathbf{x}_{m'})\log p_\theta(\mathbf{x}_m|\mathbf{z})d\mathbf{z}d\mathbf{x}_md\mathbf{x}_{m'}$$

$$= \mathbb{E}_{p_d(\mathbf{x}_m,\mathbf{x}_{m'})}\left[\mathbb{E}_{q_\phi(\mathbf{z}|\mathbf{x}_m,\mathbf{x}_{m'})}[\log p_\theta(\mathbf{x}_m|\mathbf{z})]\right].$$

## F   RESULTS ON MNIST-SVHN-TEXT WITH STANDARD DEVIATIONS

Table 3 and Table 4 are Table 1 and Table 2 with their standard deviations of three trials explicitly added. The meaning of MVTCAE (rescale) is explained in Appendix H.

## G   PERFORMANCE ON THE LATENT REPRESENTATION

Table 5 shows that coordination-based performs better as input modality increases. This is a feature of the coordination-based method, which learns to approximate the joint posterior, so the amount of information to be inferred increases as the number of modalities increases. On the other hand, the mixture-based method performs better given a single modality. The proposed method improves unimodal inference the most among the coordination-based methods.

Table 3: Cross-coherence on MNIST-SVHN-Text with standard deviations.

| MODEL | R | M | | | S | | |
| | | S | T | S+T | M | T | M+T |
|---|---|---|---|---|---|---|---|
| MVAE | $0.04_{\pm 0.01}$ | $0.29_{\pm 0.04}$ | $0.11_{\pm 0.00}$ | $0.30_{\pm 0.04}$ | $0.36_{\pm 0.08}$ | $0.29_{\pm 0.08}$ | $0.67_{\pm 0.05}$ |
| MMVAE | $0.12_{\pm 0.01}$ | $0.81_{\pm 0.00}$ | $0.99_{\pm 0.00}$ | $0.91_{\pm 0.00}$ | $0.39_{\pm 0.01}$ | $0.44_{\pm 0.02}$ | $0.41_{\pm 0.01}$ |
| MoPoE-VAE | $0.12_{\pm 0.01}$ | $0.82_{\pm 0.00}$ | $0.99_{\pm 0.00}$ | $0.94_{\pm 0.01}$ | $0.35_{\pm 0.02}$ | $0.37_{\pm 0.02}$ | $0.36_{\pm 0.02}$ |
| MMJSD | $0.09_{\pm 0.00}$ | $0.81_{\pm 0.01}$ | $0.99_{\pm 0.00}$ | $0.90_{\pm 0.00}$ | $0.39_{\pm 0.01}$ | $0.41_{\pm 0.01}$ | $0.40_{\pm 0.01}$ |
| MMJSD-PoE | $0.15_{\pm 0.03}$ | $0.12_{\pm 0.01}$ | $0.78_{\pm 0.03}$ | $0.80_{\pm 0.03}$ | $0.56_{\pm 0.01}$ | $0.73_{\pm 0.03}$ | $0.86_{\pm 0.01}$ |
| MVTCAE | $0.09_{\pm 0.00}$ | $0.23_{\pm 0.01}$ | $0.87_{\pm 0.02}$ | $0.90_{\pm 0.01}$ | $0.59_{\pm 0.01}$ | $0.78_{\pm 0.00}$ | $0.80_{\pm 0.01}$ |
| MVTCAE (RESCALE) | $0.14_{\pm 0.00}$ | $0.29_{\pm 0.02}$ | $0.89_{\pm 0.01}$ | $0.92_{\pm 0.01}$ | $0.70_{\pm 0.01}$ | $0.79_{\pm 0.01}$ | $0.82_{\pm 0.01}$ |
| **CRMVAE** | $0.15_{\pm 0.01}$ | $0.60_{\pm 0.01}$ | $0.97_{\pm 0.01}$ | $0.97_{\pm 0.01}$ | $0.73_{\pm 0.01}$ | $0.87_{\pm 0.01}$ | $0.87_{\pm 0.01}$ |

| MODEL | T | | |
| | M | S | M+S |
|---|---|---|---|
| MVAE | $0.13_{\pm 0.01}$ | $0.20_{\pm 0.07}$ | $0.22_{\pm 0.08}$ |
| MMVAE | $0.48_{\pm 0.05}$ | $0.40_{\pm 0.04}$ | $0.44_{\pm 0.05}$ |
| MoPoE-VAE | $0.49_{\pm 0.04}$ | $0.42_{\pm 0.03}$ | $0.51_{\pm 0.04}$ |
| MMJSD | $0.44_{\pm 0.02}$ | $0.37_{\pm 0.01}$ | $0.40_{\pm 0.02}$ |
| MMJSD-PoE | $0.71_{\pm 0.02}$ | $0.14_{\pm 0.02}$ | $0.74_{\pm 0.04}$ |
| MVTCAE | $0.76_{\pm 0.02}$ | $0.27_{\pm 0.02}$ | $0.79_{\pm 0.02}$ |
| MVTCAE (RESCALE) | $0.87_{\pm 0.01}$ | $0.34_{\pm 0.02}$ | $0.89_{\pm 0.02}$ |
| **CRMVAE** | $0.91_{\pm 0.02}$ | $0.71_{\pm 0.00}$ | $0.95_{\pm 0.01}$ |

Table 4: FID score of the generated SVHN images with standard deviations.

| MODEL | S→M | M→S |
|---|---|---|
| MVAE | $39.0_{\pm 4.73}$ | $36.0_{\pm 2.08}$ |
| MMVAE | $91.8_{\pm 1.46}$ | $307.9_{\pm 0.92}$ |
| MoPoE-VAE | $85.3_{\pm 2.51}$ | $312.8_{\pm 1.81}$ |
| MMJSD | $86.7_{\pm 1.94}$ | $287.0_{\pm 1.15}$ |
| MMJSD-PoE | $70.0_{\pm 1.17}$ | $141.4_{\pm 3.20}$ |
| MVTCAE | $41.6_{\pm 1.63}$ | $31.8_{\pm 0.51}$ |
| MVTCAE (RESCALE) | $32.9_{\pm 0.11}$ | $35.2_{\pm 0.35}$ |
| **CRMVAE** | $24.1_{\pm 0.71}$ | $37.3_{\pm 0.52}$ |

The right panels of Figure 5 and Figure 6 show the inference performance of PolyMNIST and Translated-PolyMNIST, respectively. It can be seen that the proposed method performs well when the number of modalities is small, while other models perform better when the number of modalities increases. There are various possible reasons for this, but the scaling of the reconstruction error term, which will be discussed in the Appendix H, might be the cause.

Table 5: Performance on the latent representation in MNIST-SVHN-Text.

| MODEL | M | S | T | M+S | M+T | S+T | M+S+T |
|---|---|---|---|---|---|---|---|
| MVAE | $0.73_{\pm 0.08}$ | $0.44_{\pm 0.02}$ | $0.30_{\pm 0.04}$ | $0.84_{\pm 0.07}$ | $0.97_{\pm 0.00}$ | $0.58_{\pm 0.08}$ | $0.97_{\pm 0.08}$ |
| MMVAE | $0.97_{\pm 0.01}$ | $0.82_{\pm 0.05}$ | $0.99_{\pm 0.00}$ | $0.89_{\pm 0.04}$ | $0.98_{\pm 0.00}$ | $0.91_{\pm 0.01}$ | $0.93_{\pm 0.05}$ |
| MoPoE-VAE | $0.95_{\pm 0.02}$ | $0.81_{\pm 0.04}$ | $0.99_{\pm 0.00}$ | $0.96_{\pm 0.03}$ | $0.98_{\pm 0.00}$ | $0.93_{\pm 0.02}$ | $0.98_{\pm 0.04}$ |
| MMJSD | $0.97_{\pm 0.01}$ | $0.82_{\pm 0.02}$ | $0.99_{\pm 0.00}$ | $0.89_{\pm 0.01}$ | $0.98_{\pm 0.00}$ | $0.91_{\pm 0.01}$ | $0.93_{\pm 0.01}$ |
| MMJSD-PoE | $0.75_{\pm 0.01}$ | $0.14_{\pm 0.02}$ | $0.94_{\pm 0.01}$ | $0.77_{\pm 0.02}$ | $0.99_{\pm 0.03}$ | $0.95_{\pm 0.03}$ | $0.99_{\pm 0.03}$ |
| MVTCAE | $0.91_{\pm 0.01}$ | $0.31_{\pm 0.02}$ | $0.99_{\pm 0.01}$ | $0.93_{\pm 0.02}$ | $0.99_{\pm 0.02}$ | $0.99_{\pm 0.00}$ | $0.99_{\pm 0.02}$ |
| MVTCAE (RESCALE) | $0.88_{\pm 0.01}$ | $0.35_{\pm 0.01}$ | $0.95_{\pm 0.02}$ | $0.90_{\pm 0.02}$ | $0.96_{\pm 0.01}$ | $0.96_{\pm 0.01}$ | $0.96_{\pm 0.02}$ |
| **CRMVAE** | $0.95_{\pm 0.01}$ | $0.74_{\pm 0.02}$ | $0.99_{\pm 0.01}$ | $0.97_{\pm 0.00}$ | $0.99_{\pm 0.01}$ | $0.99_{\pm 0.01}$ | $0.99_{\pm 0.01}$ |

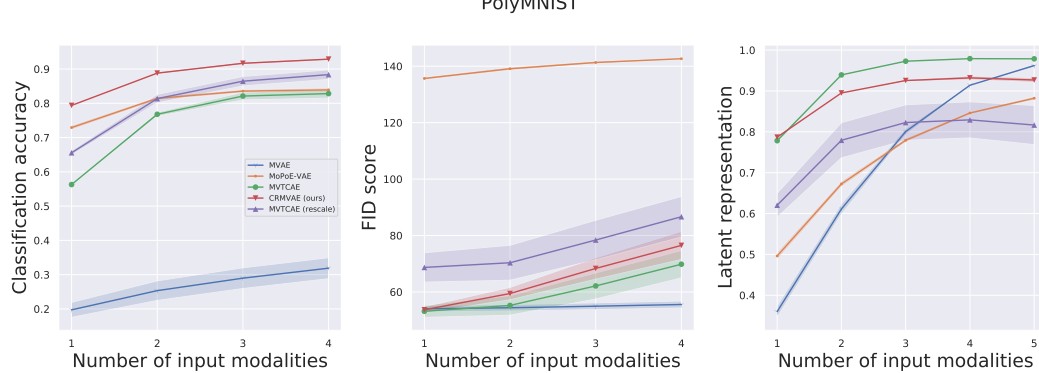

Figure 5: Performance evaluation on PolyMNIST. The left panel shows the performance of cross-coherence. The right is the performance of linear classification on the latent space given an arbitrary modality set.

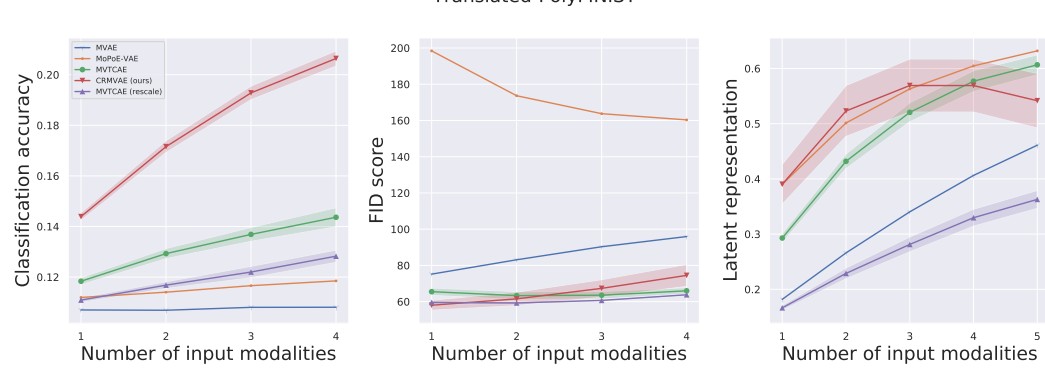

Figure 6: Performance evaluation on Translated-PolyMNIST.

## H    CONSIDERATION OF THE COEFFICIENTS OF THE RECONSTRUCTION TERM

We chose $\pi$ instead of $\frac{1}{2}$ as the coefficient in Eq. 8. However, this makes the reconstruction error term smaller than the other models by a factor of $\frac{\pi}{2}$. To check this effect, we scaled the MVTCAE reconstruction error term by multiplying it by $\frac{\pi}{2}$. Since this effect is large when the modality is three or more, we experimented with MNIST-SVHN-Text, PolyMNIST, and Translated-PolyMNIST.

The results denoted as "MVTCAE (rescale)" are the MVTCAE results aligned in scale with the reconstruction error term in CRMVAE. First, for MNIST-SVHN-Text, the performances of cross-coherence and FID are slightly better with scale, but CRMVAE is still better. The joint coherence was comparable to that of CRMVAE. However, the performance of the inferred representation is lower with scaling. This is because the coordination-based learns inference by reconstruction, and when this term is small in the overall expression, it fails to learn inference. Next, PolyMNIST and Translated-PolyMNIST improve cross-coherence performance, but as with MNIST-SVHN-Text, inference performance is poor. Compared to MVTCAE with rescaling, the performance of CRMVAE is better, and this comparison, in which the reconstruction error terms are scaled to each other, might better evaluate the effect of unimodal reconstruction included in CRMVAE.

These results show that adjusting for scale does not change the superiority of CRMVAE. In other words, the higher performance of CRMVAE is not due to scaling alone.

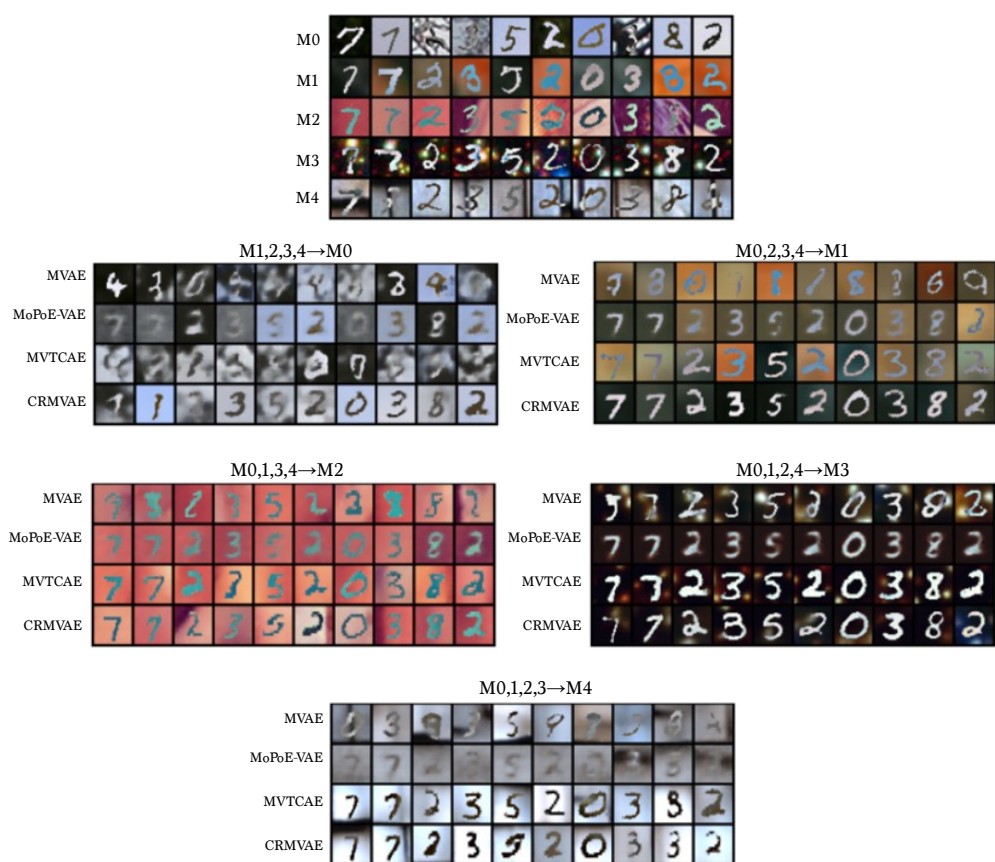

Figure 7: Cross-generation of PolyMNIST. The top panel shows examples of modalities M0 to M4 (we refer to the five different modalities as M0 to M4), and the bottom panels show the generation of a modality conditioned on remaining modalities by each model.

## I   CONDITIONAL GENERATION ON POLYMNIST AND TRANSLATED-POLYMNIST

Figure 7 and Figure 8 visualize the conditionally generated images on the PolyMNIST and Translated-PolyMNIST datasets.

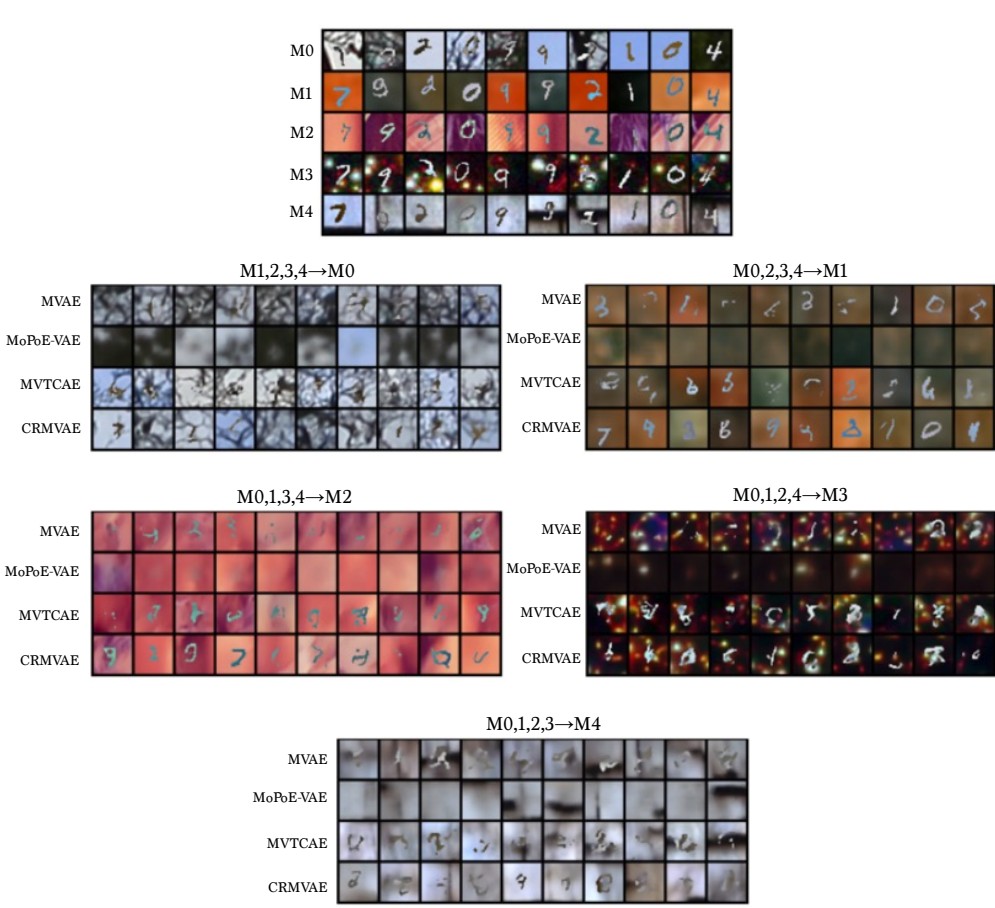

Figure 8: Cross-generation of Translated-PolyMNIST.

