# OpenReview forum: "Mitigating the Limitations of Multimodal VAEs with Coordination-Based Approach"
_ICLR.cc/2023/Conference — Submitted to ICLR 2023_

### Official Review · Reviewer_Jrda · 2022-10-25

**Confidence:** 3
**Correctness:** 4
**Technical Novelty And Significance:** 3
**Empirical Novelty And Significance:** 3
**Recommendation:** 5

**Clarity, Quality, Novelty And Reproducibility:**

The paper is generally well written. The presentation is largely clear. However, it would be clearer to summarize exactly what constitutes the method CRMVAE: Eq. (5) that replaces the reconstruction term with eq. (7), and a MoE aggregation?
Relevant prior work is cited appropriately.


**Strength And Weaknesses:**

Strengths:
- The idea to include the uni-modal reconstruction terms appears well motivated.
- The empirical performance suggests that the new approach yields better generative quality (e.g., FID scores) or cross-coherence (e. g., classification accuracy) than most previous approaches.
- The paper adds some interesting connection between MVTCAE and JMVAE and mixture-based multi-modal VAEs.

Weaknesses:
- While the suggested approach improves the cross-coherence compared to previous work, the generation quality does not improve necessarily compared to some methods.
- The algorithmic change to include the unimodal reconstruction terms is somewhat small.

Comments:
- To clarify Figure 2, do the different number of modalities mean that a different generative model is trained for different number of modalities, or is this the same generative model that has a different number of missing modalities when encoding the latents?
- Does Lemma 1 assume a MoE approximate posterior aggregation as in Lemma 2?
- Does the approach give good accuracy to classify the labels for PolyMNIST/Translated-PolyMNIST in the latent space?


**Summary Of The Paper:**

The paper suggests modifying the variational objective in mixture-based multi-modal VAEs by including both a uni-modal as well as a multi-modal reconstruction term. The authors also aim to present a unified perspective on mixture-based multimodal VAEs by showing that methods motived by a Jensen-Shannon-Divergence or Total Correlation objective can be viewed as mixture-based VAEs under some assumptions. Empirical results indicate the proposed approach improves the tradeoff of generative quality versus cross-coherence for different multi-modal VAEs.


---
Post-rebuttal comments:
Having read the other reviews and the authors' response, I feel that the submission is stil borderline.
The authors response clarified the comments that I had previously.
The presentation in the updated version has been improved.

---

**Summary Of The Review:**

The suggested approach of including both uni- and multi-modal reconstruction terms is novel as far as I am aware. It appears to be a rather small adjustment algorithmically compared to previous work yet it can yield improved performance, particularly with respect to the cross-modal coherency.

---

> ### Author Response · Authors · 2022-11-19
> **Reply to reviewer Jrda**
>
> Thank you very much for your insightful comments. We will respond to the points you raised.
>
> > To clarify Figure 2, do the different number of modalities mean that a different generative model is trained for different number of modalities, or is this the same generative model that has a different number of missing modalities when encoding the latents?
>
> The latter is correct. For each method, we trained a generative model for all modalities and then evaluated the performance of the same generative model with different numbers of missing modalities. We have added a description of the experiments in Section 6, as we did not explain them well enough.
>
> > * Does Lemma 1 assume a MoE approximate posterior aggregation as in Lemma 2?
>
> Thank you for your question. The proof of Lemma 1 makes no assumptions about the approximate posterior. We clearly stated this in the paper.
>
> > * Does the approach give good accuracy to classify the labels for PolyMNIST/Translated-PolyMNIST in the latent space?
>
> We have included in the appendix the classification results of MNIST-SVHN-Text, PolyMNIST, and Translated-PolyMNIST on latent variables. First, for MNIST-SVHN-Text, the proposed method generally performed well. On the other hand, for PolyMNIST and Translated-PolyMNIST, the proposed method performed slightly worse than the others. Since the proposed method has multiple reconstruction error terms, we normalized the reconstruction error terms according to the number of modalities (specifically, we changed the coefficient of Eq. 7 in the proposed method from $\frac{1}{2}$ to $\pi=\frac{1}{M+1}$, where M is the number of modalities), and we considered that one of the reasons was that this underestimated the reconstruction term in the overall objective. We therefore performed an experiment with the competing method, MVTCAE, scaled by the same factor to check under the same conditions and found that the proposed method performed better. We have included these considerations in the paper. Please refer to the appendix for details.
>
> > it would be clearer to summarize exactly what constitutes the method CRMVAE: Eq. (5) that replaces the reconstruction term with eq. (7), and a MoE aggregation?
>
> CRMVAE uses PoE-aggregation for approximate posterior. We added this explanation to the paper because it was not clear.

---

### Official Review · Reviewer_P3oY · 2022-10-25

**Confidence:** 4
**Correctness:** 3
**Technical Novelty And Significance:** 2
**Empirical Novelty And Significance:** 3
**Recommendation:** 5

**Clarity, Quality, Novelty And Reproducibility:**

**Clarity**: the flow and exposition of the paper are clear. However, it would be preferable to have more results (e.g. in the Appendix) for a clearer understanding of model performance in the experiments (e.g. generation when latent code is sampled from a prior, image-to-caption generation).

**Novelty**: This work analyzes the relationship between MVTCAE and MMJSD, and categorizes them as coordination-based approaches. Then proposes CRMVAE, which extends the MVTCAE by adding unimodal reconstruction terms. The authors stress that their addition is beneficial for model performance and this is confirmed in the showcased experimental results, but the introduction of these terms is backed only with an intuitive argument, and more empirical results would be helpful to see if limitations of existing multimodal VAEs are fully addressed (see point 2 Weaknesses).

**Reproducibility**: The authors did not share their code and the paper alone does not provide sufficient details, which makes it hard to reproduce their results.

**Quality**: There are some aspects of high-quality work, such as the inclusion of a variety of experimental settings, but also some aspects in which quality could be improved, such as reproducibility.


**Strength And Weaknesses:**

**Strengths**:
- The paper investigates a relevant problem, trying to overcome the limitations of existing multimodal VAEs.
- The paper is well-written and well-structured.
- The Experiments section shows the authors have made an effort to evaluate their proposed method on a wide range of datasets, and empirical results indicate a clear improvement over previous methods.

**Weaknesses**:

- The main contribution of the authors when proposing the CRMVAE is introducing unimodal reconstruction terms in the objective. However, the rationale behind this choice seems to be rather based on intuition and lacks theoretical grounding, leaving some open questions. It is e.g. unclear why optimizing for unimodal reconstruction of each modality, i.e. reconstruction of each modality given the latent code inferred from the corresponding unimodal encoder, would improve coherence for cross-generation.
- The experimental results are extensive in the sense of models being evaluated on a wide range of datasets, but there would be additional results that would give important insights on whether certain current limitations of multimodal VAEs are truly overcome. For example, how do the models compare on generation performance, when the joint latent code is sampled from a prior? This is an important aspect when trying to learn a joint distribution of all modalities [1-2], and I think coherence across generated modalities in this setting would show if bringing unimodal and joint posteriors closer together effectively leads to encoding shared content effectively in the latent space.
- From my understanding it is unclear if the equality in section 3.4 below equation 7, central to deriving the ELBO, is justified for the considered setting. Could the authors elaborate on that? Also, it seems like the assumption could be stated more clearly: Are the integrals actually necessary for the left-hand side, or is it equivalent to assuming that the joint posterior is equal to each unimodal posterior? Can the assumption be stated in terms of a (conditional) independence relation?
- Another important point where this work can in my opinion be improved is reproducibility. Being quite familiar with the mentioned models and experimental settings, I would have liked to replicate some of the results shown in the paper, but the authors did not share their code, which makes it hard to do so. In addition, I tried to reproduce the results for MVTCAE in the CUB Image-Captions experiment, using the public repo for the original paper, but I was not able to obtain the same performance shown in the paper for cross-generation, despite using the same settings. I think it would be helpful to see the code to understand the implementation better and reproduce the results.
- [Minor] I believe a logarithm is missing in the RHS of the equation in Proof 1.

**References**
- [1] Yuge Shi, et al.: Variational Mixture-of-Experts Autoencoders for Multi-Modal Deep Generative Models. NeurIPS 2019.
- [2] Thomas M. Sutter, et al.: Multimodal Generative Learning Utilizing Jensen-Shannon-Divergence. NeurIPS 2020.

**Summary Of The Paper:**

The paper deals with multimodal VAEs. In particular it tries to mitigate the current limitations of models in this class. After highlighting the shortcomings of mixture-based approaches, which have been uncovered in previous work, the authors investigate the relationship between two previously proposed product-based multimodal VAEs, namely MVTCAE and MMJSD, and categorize them as coordination-based approaches. Coordination-based multimodal VAEs assume a product-of-experts joint posterior to have high-quality generated samples, and introduce specific terms in the objective to bring unimodal and joint encoders closer together, to obtain coherence between modalities. The authors propose CRMVAE, by extending the MVTCAE to include unimodal reconstruction terms in the objective, and show empirical results in multiple experiments indicating an overall improvement over existing approaches in terms of coherence and quality of generation.

**Summary Of The Review:**

The paper is well-written, deals with a relevant problem, and presents promising empirical results. The authors propose a model in the class of coordination-based multimodal VAEs that extends MVTCAE, by adding unimodal reconstruction terms in the objective. However, behind this addition, there only seems to be an intuitive argument. While empirical results are promising, reproducibility can be improved, and more empirical results could give a clearer picture on whether limitations of existing multimodal VAEs are fully addressed.

---

> ### Author Response · Authors · 2022-11-19
> **Reply to reviewer P3oY**
>
> Thank you for your detailed review and many insightful comments. We will respond to your points.
>
> > The main contribution of the authors when proposing the CRMVAE is introducing unimodal reconstruction terms in the objective. However, the rationale behind this choice seems to be rather based on intuition and lacks theoretical grounding, leaving some open questions.
>
> Thank you for asking about a very essential point. Our proposed method is indeed intuitive, but it is related to coordination-based methods and PoE aggregation. First, in the coordination-based method, the inference of arbitrary modalities tends to be ignored in the learning process since there is no cross-generation among all modalities as in the mixture-based method. Therefore, we perform unimodal inference reconstruction to ensure that all unimodal inferences are properly inferred and generated.
> Next, let us discuss the relationship with PoE. since PoE represents joint posterior as a product of unimodal posterior, it should be possible to learn inferences from any modality simply by learning inferences of joint posterior. However, as mentioned above, arbitrary inferences tend to be ignored in the learning process, so unimodal reconstruction is required. Conversely, if unimodal inference can be trained properly, PoE can approximate inference from arbitrary inputs thanks to its product property. We also showed that the reconstruction term from arbitrary modalities could be derived by transforming the unimodal reconstruction term  (Equation 9). Thus, learning unimodal reconstruction leads to improvements in both inference and generation from arbitrary modality inputs. We have added such a discussion to the section on the proposed method since it was not written in the previous version.
>
> > how do the models compare on generation performance, when the joint latent code is sampled from a prior?
>
> We experimented with joint coherence in MNIST-SVHN-Text with reference to previous studies. As a result, the performance was not as good as the previous study, but the proposed method had some advantages. We believe this is mainly due to the difference in the evaluation method of text modality.
>
> > From my understanding it is unclear if the equality in section 3.4 below equation 7, central to deriving the ELBO, is justified for the considered setting. Could the authors elaborate on that? Also, it seems like the assumption could be stated more clearly: Are the integrals actually necessary for the left-hand side, or is it equivalent to assuming that the joint posterior is equal to each unimodal posterior? Can the assumption be stated in terms of a (conditional) independence relation?
>
> To answer this comment, we have made several revisions to the paper.
> First, the notations were corrected. Specifically, I separated the notation into $q_{\phi}$ for joint posterior and $q_{\phi_m}$ for unimodal posterior. This makes it easier to understand whether we are discussing joint or unimodal posterior. Next, we introduce Lemma 3 and its proof, which shows that optimizing MVTCAE encourages $q_{\phi}(z|x_m)$ and $q_{\phi_m}(z|x_m)$ to get closer. This holds for any approximate posterior. In other words, it can be shown that $q_{\phi}(z|x_m)\approx q_{\phi_m}(z|x_m)$ by the optimization of MVTCAE, and for any distribution, this optimization brings the equality hypothesis, which you pointed out, closer to being valid.
> Then we use the joint inference $q_{\phi}(z|X)=p(z)\prod_m q_{\phi_m}(z|x_m)$ with PoE, and unimodal inference is also possible by setting $q_{\phi}(z|x_m)=p( z)q_{\phi_m}(z|x_m)$ using PoE instead of $q_{\phi}(z|x_m)$, which makes $q_{\phi}(z|x_m)=q_{\phi}(z|x_m)$ and the equality you pointed out is valid.
>
> > I think it would be helpful to see the code to understand the implementation better and reproduce the results.
>
> We have attached our implementation of the model construction and training as supplemental material. We hope this will improve your evaluation of the reproducibility of this study.
>
> > * [Minor] I believe a logarithm is missing in the RHS of the equation in Proof 1.
>
> The points you raised have been corrected.

---

> > ### Comment · Reviewer_P3oY · 2022-12-07
> > **Thank you for the reply**
> >
> > Dear authors,
> >
> > thank you for the reply.
> >
> > I appreciate the effort in explaining the rationale behind the proposed objective. The introduction of unimodal reconstruction terms still seems to be primarily based on intuition and lacks theoretical grounding.
> >
> > An evaluation of unconditional generation performance (i.e., when the joint latent code is sampled from a prior) is still missing, despite it being an important and widely-used criterion for the evaluation of (multimodal) VAEs.
> >
> > Thank you for elaborating on the matching of variational posteriors. Even though the matching sounds intuitive, the response still does not make it sufficiently clear whether Equation (8) holds in a general or whether it requires further assumptions, such as $q_{\phi_m}(z | x_m) = q_{\phi}(z | x_m)$ or $q_{\phi_m}(z | x_m) = q_{\phi}(z | X)$. Moreover, the paper changed in several ways in the context of Equation (8), but the changes still do not make the assumptions sufficiently clear. I believe that a clearer statement of the assumptions would improve the clarity of the work without limiting its potential impact.
> >
> > With these points in mind, I choose to keep my score.

---

### Official Review · Reviewer_venE · 2022-10-29

**Confidence:** 3
**Correctness:** 4
**Technical Novelty And Significance:** 2
**Empirical Novelty And Significance:** 2
**Recommendation:** 5

**Clarity, Quality, Novelty And Reproducibility:**

The paper is very clear. Novelty is not sufficiently argued for (I am referring to the weighted reconstruction loss).

**Strength And Weaknesses:**


The paper is very well written and easy to follow. I have the following comments and suggestions.

Comments:
- How is the unimodal posterior $q_\theta(z|X)$ computed and sampled from? The main motivation of mixture models is to be scalable, i.e., they can be used for large number of modalities just as effectively. The equations described in (5) and (7) look like something that is not going to scale very well with the number of modalities. Is it possible to show how the approach scales as the modality increase, empirically?
- The paper reads more like a survey. It would have been much better if the Sections from Section 2. up to Subsection 3.3 are condensed and the contributions, Section 3.3 and Section 3.4, are expanded upon.
- Related works is not specialized to this work-- there is a paragraph about transformer based multimodal models.

Suggestions:
- Please include what $H(\cdot)$ actually is, this is below Eq(4). It is much better to define it there than forcing the reader to scroll through the reference.
- Please improve result reporting in Table-1. The abbreviated dataset description and their categorization is not clear.

**Summary Of The Paper:**

The paper describes recent development in Multimodal VAE in extensive details.
Specifically, versions of the ELBO where the latent models , based on subset of modalities, are forced to be similar to
the latent model that is estimated with all the modalities is explained as Product of Experts like model. The paper goes on
to propose a weighted sum of reconstruction loss together with Product of Experts like posterior model.


**Summary Of The Review:**

The paper is an informative piece of work on multimodal VAE. Although, it reads like a survey paper with added suggestions. My main concern is the scalability of the approach which mixture models are meant to be efficient on.

---

> ### Author Response · Authors · 2022-11-19
> **Reply to reviewer venE**
>
> Thank you for your very helpful review and suggestions. We would like to respond to your points.
>
> > How is the unimodal posterior computed and sampled?
>
> We apologize for the lack of explanation. The proposed method uses an approximate posterior based on PoE-aggregation. That is, each unimodal posterior is parameterized by DNN, and the posterior given arbitrary inputs can be computed analytically by their product. Thus, $q(z|X)$ in equations (5) and (7) are computed by PoE-aggregation, and scalability is maintained with respect to the number of modalities. We have modified the explanation in the coordination-based and proposed methods to clarify this point.
>
> > The paper reads more like a survey. It would have been much better if the Sections from Section 2. up to Subsection 3.3 are condensed and the contributions, Section 3.3 and Section 3.4, are expanded upon.
>
> Thank you very much for your suggestions.
> First, we believe that part of the contribution of this study is to show that MVTCAE and MMJSD can be viewed as a unified coordination-based framework and that they can address the limitation, as pointed out in the first half of Section 3.
> However, as you pointed out, the structure of this paper was not appropriate, especially in sections 2 and the first half of 3, and it is true that our paper is like a survey. Therefore, we have changed to the structure of this paper.
> Section 3.1 is placed at the end of Section 2, Section 3.2 and Section 3.3 are placed in Section 3, and Section 3.4, which describes the proposed method, is now an independent Section 4.
> We have shortened the discussion of existing studies, enriched the newly introduced proofs in Section 3, and enriched the description of the proposed method in Section 4.
> We also separated the discussion of related studies in Section 3.5 and the proposed method in Section 5.
>
> > Related works is not specialized to this work-- there is a paragraph about transformer based multimodal models.
>
> As you suggested, we have removed the description of related studies on transformers in Section 3.5.
>
> > Please include what H. actually is, this is below Eq(4). It is much better to define it there than forcing the reader to scroll through the reference.
>
> H stands for conditional entropy. We have added this explanation as you suggested.
>
> > Please improve result reporting in Table-1. The abbreviated dataset description and their categorization is not clear.
>
> We have added a specific explanation of how to read Table 1 in this caption.

---

### Official Review · Reviewer_DC9h · 2022-11-04

**Confidence:** 1
**Clarity, Quality, Novelty And Reproducibility:** The paper is well written, and the pr…
**Correctness:** 3
**Technical Novelty And Significance:** 3
**Empirical Novelty And Significance:** 3
**Recommendation:** 8

**Strength And Weaknesses:**


Strength:

- The authors point out why these existing coordinate-based models perform poorly on cross-model generation, and propose a novel model to fix the issue.
- The proposed model can mitigate the limitations in multimodal VAEs and performs well in both cross-coherence and generation quality.


**Summary Of The Paper:**

It is challenging to infer a joint representation from arbitrary subsets of multimodalities, and the state-of-the-art approaches (mixture-based multimodal VAEs) attempt to accomplish this by training to generate all modalities from a joint representation inferred from missing modalities, but the quality of modality generation is lower than that of unimodal VAEs, and this limitation is theoretically unavoidable.  Therefore, the authors propose a coordination-based model that brings the representation inferred from each modality closer to that inferred from all modalities. Experiments with diverse and challenging datasets show the advances of the proposed method.

**Summary Of The Review:**

Unfortunately, this paper lies outside of my field of expertise, therefore, my comment may be biased. Area chairs are suggested to seek opinions from other reviewers.

---

> ### Author Response · Authors · 2022-11-19
> **Reply to reviewer DC9h**
>
> Thank you for your high evaluation of our paper.
> We have revised our paper to address the points mentioned in the overall comments, and we would be happy if you would review it.

---

### Author Response · Authors · 2022-11-19
**Thank you for your valuable reviews**

First, we would like to thank all reviewers for their thoughtful and informative reviews. We have made the following corrections to this paper Please check our responses to each reviewer for specifics.

- The structure of the paper has been changed to make the proposal part easier to understand.
- The description of the part related to the proposed method has been changed so that the contribution can be better understood.
- We have added one more Lemma and proof for the coordination-based model.
- We have added to the appendices additional experiments on the performance of inference to the latent space.
- We have added an experiment to evaluate the performance of joint coherence in MNIST-SVHN-Text.
- We explained that we normalize the reconstruction term in the proposed method because we are concerned that the value of the reconstruction term may become too large in the objective, and added an experiment on this effect in the appendix.
- We have corrected some notations and typographical errors.
- Other general lack of explanation has been corrected.
- We have attached an implementation involving algorithms and experiments.

---

### Decision · Program_Chairs · 2023-01-20

**Decision:**

Reject

**Justification For Why Not Higher Score:**

Overall, the reviewers all agreed on borderline rejection (the reviewer with score of 8 was not familiar with this field and lacked thorough comments) due to the split focus on a unified understanding combined with a small contribution, as well as several limitations in the performance (some that came out after rebuttal).

**Justification For Why Not Lower Score:**

N/A

**Metareview: Summary, Strengths And Weaknesses:**

This paper tackles the problem of learning multi-modal VAEs that can jointly represent modalities. The paper first analyzes the limitations of current mixture-based methods and studies product-based methods which can be categorized as coordination-based approaches. The paper then proposes a uni-modal reconstruction loss to account for the lack of cross-generation terms which are known to produce a looser bound when performing ELBO optimization. Results are shown across a number of datasets, showing improvements in generation quality and coherence.

  Overall, the reviewers appreciated the importance of the problem setting, clarity of the paper (though its focus was mostly on the connections between various methods rather than the proposed method), and the overall empirical performance. The main weaknesses that reviewers agreed on is the significance/impact of the contribution (venE, P3oY), main (non-theory-based) motivation of the approach (venE, P3oY), and in some cases empirical generation performance (Jrda). The authors rebutted some of these claims, but ultimately the reviewers were still very much borderline tending towards rejection. Indeed, some of the additional experiments (e.g. label classification results and generation when the joint latent code is sampled from a prior) showed that the additional reconstruction term presents some aspects (e.g. additional normalization) which could be problematic.

Considering that all of the reviewers had some concerns about the level of contribution of the paper (discounting the strongest score which indicated lack of expertise in the area), I recommend that the paper and approach be significantly improved and submitted to a future venue, and not accepted as is. For example, the paper mixes some interesting insights (unified coordination based framework) and then proposes a relatively small contribution. If the latter is the focus, then this should be beefed up, adding more insights and motivation for the method in addition to more convincing and thorough experimental results (including improving performance on some of the experiments suggested by the reviewers).